# Haploidentical Stem Cell Transplantation in Lymphomas—Expectations and Pitfalls

**DOI:** 10.3390/jcm9113589

**Published:** 2020-11-07

**Authors:** Jacopo Mariotti, Stefania Bramanti, Armando Santoro, Luca Castagna

**Affiliations:** 1Bone Marrow Transplant Unit, Humanitas Clinical and Research Center, 20089 Rozzano, Italy; stefania.bramanti@humanitas.it (S.B.); luca.castagna@humanitas.it (L.C.); 2Humanitas Clinical and Research Center, Humanitas University, 20089 Rozzano, Italy; armando.santoro@humanitas.it

**Keywords:** haploidentical transplantation, lymphoma

## Abstract

T-cell replete Haploidentical stem cell transplantation (Haplo-SCT) with Post-transplant cyclophosphamide (PT-Cy) is an emerging therapeutic option for patients with advanced relapsed or refractory lymphoma. The feasibility of this platform is supported by several retrospective studies showing a toxicity profile that is improved relative to umbilical cord blood and mismatched unrelated donor (UD) transplant and comparable to matched unrelated donor transplant. In particular, cumulative incidence of chronic graft-versus-host disease (GVHD) is reduced after Haplo-SCT relative to UD and matched related donor (MRD) transplant thanks to PT-Cy employed as GVHD prophylaxis. This achievement, together with a similar incidence of acute GVHD and disease relapse, results in a promising advantage of Haplo-SCT in terms of relapse-free/GVHD free survival. Unmet needs of the Haplo-SCT platform are represented by the persistence of a not negligible rate of non-relapse mortality, especially due to infections and disease relapse. Future efforts are warranted in order to reduce life-threatening infections and to employ Halo-SCT with PT-Cy as a platform to build new immunotherapeutic strategies.

## 1. Introduction

Allogeneic hematopoietic stem cell transplantation (Allo-HSCT) represents a potential curative option for patients with refractory or relapsed lymphomas thanks to the immune-mediated graft-versus-lymphoma (GVL) effect. The existence of a GVL effect is supported first by the evidence of a reduced incidence of relapse for patients with Hodgkin (HL) and non-Hodgkin (NHL) lymphoma after Allo-HSCT (ranging between 6% and 29%) relative to autologous transplant (ranging between 35% and 69%) [1,2,3,4,5]. The GVL effect was more convincingly demonstrated by resolution of residual or progressive disease after Allo-HSCT with withdrawal of immunosuppression or donor lymphocyte infusion (DLI). In particular, disease response has been achieved in about 30–56% of patients with HL [6,7,8], 40–56% of diffuse large B cell (DLBCL) NHL [9,10], 75–90% for follicular NHL (FL) [11]. Unfortunately, the benefits of a GVL effect have been jeopardized by the occurrence of transplant complications resulting in a high rate of non-relapse mortality (NRM).

The development of new drugs for relapsed/refractory lymphomas has challenged the role of allogeneic transplant in this setting. Therapeutic chances comprise monoclonal antibodies [12], check-point inhibitors [13,14] chimeric-antigen-receptor (CAR)-T cells for specific histological subtypes [15,16]. Despite the high overall response rates (ORR) achieved with new drugs, long-term outcome is still a matter of debate: only 9 out of 102 patients were apparently cured at 5-years by brentuximab in relapsed/refractory (R/R) HL [17]; median duration of response was 16.5 months for R/R HL patients treated with pembrolizumab [18] and 2-year progression free survival (PFS) was 39% in the ZUMA-1 trial for patients with DLBCL treated with CAR-T [19]. Due to the short follow-up of these studies and the lack of principle of representing a curative procedure, allo-HSCT still represents the most powerful curative tool for patients with R/R lymphomas, even if the benefits in terms of OS and PFS are hampered by a not negligible risk of NRM. Therefore, choosing the right timing and integrating new therapeutic options with Allo-HSCT is a matter of debate [20].

According to the recommendations of the European Society for Blood and Marrow Transplantation (EBMT) [21], Allo-HSCT from matched related or unrelated donor is the standard of care for lymphomas relapsing after auto-HCT. Unfortunately, HLA-identical donors, both related and unrelated, are not available for all patients and alternative donors (mismatched unrelated donor, haploidentical donor and umbilical cord blood) are searched. In the last ten years, a growing numbers of Allo-HSCT have been performed using haploidentical donor without T-cell depletion. Although several T-cell replete platforms are available with different graft-versus-host-disease (GVHD) prophylaxis, the most frequently used in US and in Europe is represented by high dose post-transplantation cyclophosphamide (PT-Cy), as pioneered by the John Hopkins and Seattle groups [22]. In the pivotal study, using a truly non-myeloablative conditioning (NMAC) regimen, the feasibility and tolerability of this platform was confirmed with low incidence of GVHD and non-relapse mortality (NRM) [22]. This achievement has contributed to the worldwide extension of haploidentical transplant (Haplo-SCT) with PT-Cy, with different levels of conditioning intensity and graft source [23,24]. Given the widespread use of this platform, this review aims at summarizing the current evidences concerning the efficacy and debated issues/unmet needs of the Haplo-SCT with PT–Cy platform (from now on called Haplo-SCT, unless differently specify) in the contest of relapsed/refractory lymphoma.

## 2. What Is the Role of Haplo-SCT vs. Other Donor Types in Hematological Malignancies?

Two large meta-analysis comparing Haplo-SCT relative to other donors’ transplants for patients with any hematologic malignancy have recently been published and comprise most of the published reports since 2013. In the first one, Meybodi et al. [25] compared the outcomes of 1410 Haplo-SCT with those of 6396 matched related donor (MRD) transplants, while in the second one Gagelmann et al. [26] analyzed a cohort of 22,974 patients receiving either a Haplo-SCT or a MRD or a matched-unrelated (MUD) or a mismatched unrelated (MMUD) donor transplant (Table 1). Main diseases were represented by acute leukemias and lymphomas. Relative to MMUD, Haplo-SCT had a superior outcome in terms of overall survival (OS) (hazard ratio (HR): 0.79, 95% confidential intervals (CI): 0.65–0.97) due to a reduced risk of NRM. This was mainly due to a reduced risk of acute and chronic GVHD (Table 1). Haploidentical and MUD transplants had comparable results in terms of OS (HR: 1.06, 95% CI: 0.96–1.18) but this was the result of opposite effects. On one side, Haplo-SCT had lower incidence of both acute and chronic GVHD resulting in reduced NRM (HR: 0.75, 95% CI: 0.61–0.92) and improved GVHD/relapse free survival (GRFS). On the other side, relapse risk was higher after Haplo-SCT compared to MUD transplants (Table 1). Both studies agreed on the fact that Haplo-SCT had an inferior OS (HR. 1.17–1.18) compared with MRD transplant, even if the difference was statistically significant only in one report. Worse outcome with Haplo-SCT was mainly related to a higher risk of NRM (HR: 1.20–1.36), probably due to increased risk of infections and delayed immune reconstitution according to Meybodi et al. [25] or to an increased risk of grade II-IV but not III-IV, acute GVHD (aGVHD) in the report by Gagelman et al. [26]. Notably, GRFS did not differ between the two donor types due to the fact that the Haplo-SCT platform was endowed by a 50% lower risk of chronic GVHD (cGVHD) with a similar relapse incidence. Put together, these two large metanalysis suggest that Haplo-SCT represent a favorable option compared with MMUD, has a better toxicity profile relative to MUD but has still a worse outcome compared with MRD transplants.

## 3. What Is the Role of Haplo-SCT vs. Other Donor Types in Lymphomas?

There are three main retrospective analyses focusing only on patients with lymphomas that were published by the Center for International Blood and Marrow Transplant Research (CIBMTR) [27,28,29]. In the first seminal report, Kanate et al. [27] compared the outcomes of 185 patients undergoing Haplo-SCT with those of 732 recipients of unrelated donor (UD) transplant with or without antithymocyte globulin (ATG). Haplo-SCT had similar results in terms of OS, progression-free survival (PFS) and NRM relative to UD transplants (Table 1) but was surprisingly more advantageous relative to UD both in terms of grade III-IV acute and chronic GVHD (HR: 0.17–0.27). Unfortunately, GRFS was not analyzed in this report. Gosh et al. [28] documented similar outcomes between Haplo (*n* = 180) and MRD (*n* = 807) transplants in terms of OS and PFS, while Haplo-SCT was hampered by significantly higher risk of NRM and aGVHD, as reported also in the metanalysis by Meybody and Gagelmann [25,26]. Consistent with Kanate et al. [27], Haplo-SCT was endowed by a strikingly lower incidence of cGVHD compared with MRD transplants (HR: 0.06; 1-year cGVHD: 12% vs. 45%, *p* < 0.001) but again data on a survival endpoint such as GRFS was unfortunately not available. More recently, Fatobene et al. [29] compared the outcome of 526 Haplo-SCTs with 214 umbilical cord blood (UCB) transplants (approximately 60–70% NHL) and found a significant advantage of Haplo-SCT in terms of OS and PFS due to a reduced NRM rate (Table 1). Moreover, the incidence of grade II-IV aGVHD (20% vs. 40%, *p* < 0.001) and cGVHD (11% vs. 17%, *p* = 0.04) were lower after Haplo-SCT relative to UCB transplants when the graft source was represented by bone marrow (BM) cells but similar when peripheral blood stem cells (PBSC) were used.

Even with the limits of retrospective registry-based studies due to the high heterogeneity in terms of disease subtypes, conditioning regimens and graft sources, all these analysis (Table 1) strongly support that Haplo-SCT with PT-Cy should be considered an acceptable option for patients with lymphoma when a MRD or MUD is not available, as recently confirmed by a position statement of the EBMT [30]. In particular, Haplo-SCT is more advantageous relative to MMUD and UCB transplants, while outcomes are very similar to MUD and MRD transplants. Moreover, given the shorter timing to identify a Haplo donor relative to a MUD, Halpo-SCT may be advantageous when disease control is a major concern. In addition, relative to the search of an UD, the availability of a haploidentical family donor has the potential benefits of lowering costs [31] and providing higher chances of transplants for patients of underrepresented minorities in international registries. Reduced incidence of cGVHD did not influence an increased risk of disease relapse suggesting that the GVT effect after Haplo-SCT is independent of chronic GVHD. Because the lower incidence of cGVHD might be due to the fact that bone marrow (BM) was the predominant graft source in Haplo-SCT, recent studies, either retrospective [32] or prospective [33], have shown that the PT-Cy platform brings similar results in terms of cGVHD and GRFS among different donor types.

## 4. Results of Haplo-SCT in HL

Three main registry-based studies analyzed the outcomes of HL patients after Haplo-SCT and performed a comparison with other donor sources. Relative to UD transplants, Kanate et al. [27] performed a sub analysis within the aforementioned study and reported similar OS and PFS between the two platforms: 3-year OS was 82% for Haplo-SCT vs. 83–79% after UD with/without ATG and 3-year PFS was 63% vs. 61–60%, respectively. Consistently, Martinez et al. [34] for the Lymphoma working party of the EBMT group and Ahmed et al. [35] for the CIBMTR described similar outcomes between Haplo-SCT and MRD transplants for HL patients in terms of OS and PFS (Table 1). A trend for higher NRM after Haplo-SCT was described in both reports. This was probably due to a higher incidence of grade II-IV aGVHD, at least in the CIBMTR analysis. Results from other single or multi-center studies concerning the Haplo-SCT platform in HL patients are summarized in Table 2. Long-term results of these reports are quite promising: 3-/4-year OS ranges between 54% and 77%, PFS is 38–66%, NRM rate is acceptable ranging between 4% and 26%, that is in line with that reported after MRD or MUD transplants.

The CIMBTR report found that Haplo-SCT was endowed by a significantly lower risk (HR: 0.45) of cGVHD relative to MRD transplants. This strikingly different incidence of cGVHD in HL patients translated in a significant improvement in terms of GRFS or extensive cGVHD/relapse free survival (CRFS). These composite endpoints are very useful to evaluate the long-term efficacy of a transplant procedure since they estimate the frequency of patients free of disease and of immunosuppression, thus reflecting potentially cured subjects experiencing a better quality of life. In details, Martinez et al. [34] found that the 1-y CRFS was 43% after Haplo-SCT, relative to 28% after MRD transplants. Consistently Gauthier et al. [36] for the Francophone Society of Bone Marrow Transplantation and Cellular Therapy described a 3-year GRFS of 37% after Haplo-SCT vs. 15% for MRD transplants (Table 1 and Table 2). Such interesting low rates of cGVHD and promising results in terms of GRFS are confirmed by other single or multi-center non registry based studies summarized in Table 2.

Another important and unexpected observation coming both from the EMBT and the CIBMTR studies is that the risk of HL relapse is significantly reduced after Haplo-SCT relative to MRD transplants (HR: 0-69-0.74, Table 1). This result is confirmed by other single center reports. Burroughs et al. [37] for the Fred Hutchinson Institute described a 2-year relapse incidence (RI) of 40% for haploidentical transplants (Table 2) relative to 56% for MRD and 63% for MUD transplants, resulting into an improved 2-year PFS (51% vs. 29% vs. 23%). Consistently, we have described [38] that 3-year RI was significantly lower after Haplo-SCT (13%, Table 2) relative to HLA identical transplants (62%) and that this was independent of pre-transplant disease status. Taken together, these results suggest a preserved or even stronger immunological activity after haploidentical transplant. Given the importance of cytokines and chemokines interactions at the level of HL microenvironment providing the Reed Stenberg cells with an anti-apoptotic and tumor-escape phenotype [39], we may speculate that the enhanced GVL effect observed after Haplo-SCT may be due to a more potent graft versus microenvironment activity.

To summarize, Haplo-SCT is an effective strategy for HL patients, its efficacy is comparable to MRD and MUD transplants and hold promises to be more advantageous in terms of reduced cGVHD and RI. Due to the lack of prospective clinical trials, a recent EBMT consensus [33] recommended a haploidentical donor transplant within the PT-Cy platform only when a MRD or a MUD is not available or when there is an urgency to find a donor if a MRD is not present. At our center, our policy is to employ a haploidentical first or second degree relative when a MRD is not available, without looking for a MUD. Without a prospective clinical trial, it is impossible to recommend a haploidentical donor over a MRD to reduce the incidence of relapse given the higher NRM rates reported after Haplo-SCT. Because NRM after Haplo-SCT is not related to cGVHD, future efforts are warranted to reduce mortality mainly due to infections or aGVHD.

## 5. Risk Factors and Unmet Needs of Haplo-SCT for HL

Similar to other donor sources, the most commonly described risk factors affecting the outcome of patients undergoing Haplo-SCT comprise pre-transplant disease status (either CR vs. not in CR or CR/PR vs. active disease), patient performance status (Karnofsky) and hematopoietic stem cell transplantation comorbidity index (HCT-CI) (either 0 vs. >0 or <3 vs. ≥3) (Table 3). In 2013, Raiola et al. [40] (Table 2) reported encouraging outcomes in 25 R/R HL patients receiving BM Haplo-SCT after the Baltimore non-myeloablative conditioning regimen (NMAC) and described pre-transplant disease status as the main variable affecting PFS (Table 3). In 2017, Castagna et al. [41] described the outcome of 62 advanced HL patients receiving a BM graft after either the Baltimore NMAC or a reduced-intensity conditioning (RIC) comprising higher dose of cyclophosphamide and thiotepa (Table 2). The intensification of the conditioning was planned to overcome the relevant rate of lymphoma relapse in the original study by Luznkik et al. [22] with the aim of testing whether stronger conditioning could ameliorate disease control. Results were consistent with those by Raiola [40] both in terms of main outcomes (Table 2) and for identifying pre-transplant disease status as the main predictor not only of PFS but also of OS, NRM and RI (Table 3). The Spanish experience (Table 2) was reported by Gayoso et al. [42], where the original Baltimora NMAC was modified into a RIC regimen with total body irradiation (TBI) replaced by busulfan at a non-myeloablative dose. Again pre-transplant disease status was the main independent prognostic variable (Table 3). In a recent update of the Genoa experience [43] comprising 41 patients, disease status assessed by fluordeoxyglucose- positron emission tomography (FDG-PET) scan (Deauville score ≥ 4) and HCT-CI ≥ 3 were the only factors associated with a worse outcome in terms of RI, PFS and GRFS (Table 3). Moreover, HCT-CI ≥ 3 was associated with increased NRM and lower OS. This study underlines the importance of achieving a deeper level of disease response before transplant to ameliorate post-transplant outcome. New drugs have shown promising results: brentuximab was associated with improved outcome in patients receiving Allo-HSCT [44]; checkpoint inhibitors (CPI), such as nivolumab, resulted in a lower chance of relapse after transplant compared with subjects not treated with CPI before transplant either from our experience (0% vs. 20%) [45] and from the John Hopkins group (PFS 90% vs. 65%) [46]. Employment of CPI was associated with increased risk of GVHD after HLA identical Hallo-SCT [47]. Such increased risk was not confirmed after Haplo-SCT employing PT-Cy for GVHD prophylaxis [45,46] suggesting that this platform is capable of reducing post-transplant alloreactivity without losing GVT response.

Choice of the better graft source, either bone marrow (BM) or peripheral blood stem cells (PBSC) for patients receiving Haplo-SCT is still matter of debate, with retrospective analysis showing either an increased risk of GVHD [48,49] with PBSC or no change [50,51]. In two different reports focused on HL, we have described [41,52] that PBSC may be preferable to BM cells because this graft source is associated with enhanced OS, PFS and GRFS. This effect is probably related to a lower chance of relapse (Table 3) mediated by the higher lymphocyte content of PBSC relative to BM. This potential advantage of PBSC for HL in terms of reduced relapse rate without increasing NRM due GVHD needs to be confirmed in a randomized clinical trial.

Choice of the best haploidentical donor may represent another strategy to improve the outcome of Haplo-SCT. In a recent publication from a multicenter retrospective analysis we have shown that choice of a younger haploidentical donor results in lower chance of aGVHD and reduced NRM, while a sibling should be preferable to a parent donor in terms of relapse incidence and PFS and a father donor is better than the mother both for OS and PFS [53].

The impact of the conditioning regimen on the outcome of transplanted patients with HL has been a matter of debate. Sureda et al. [54], in a prospective clinical trial by GEL/TAMO and EBMT comprising 46 subjects with HL receiving a MRD or MUD transplantation, described a promising 1-year NRM rate of 15%, that is lower than that described after MAC. This finding was consistent with a previous retrospective report by the EBMT [55] in HLs identifying a significantly lower 1-year NRM after RIC (23%) relative to MAC (46%) regimens. On the contrary, a more recent retrospective analysis by the EBMT lymphoma working party [56] analyzed a cohort of 312 HL recipients of MAC or RIC Allo-HSCT and described no significant increase of NRM after MAC, that was indeed advantageous in terms of reduced RI. Ciurea et al. [57] recently analyzed the effect of two variants of RIC regimens on the outcome of 15 refractory HL with active disease at the time of transplant. Indeed, an intensified RIC employing melphalan 140 mg/mq instead of melphalan 100 mg/mq was associated with a lower RI and improved PFS. A recent retrospective analysis by the CIBMTR [58] compared three different RIC regimens (fludarabine/busulfan vs. fludarabine/melphalan 140 mg/mq vs. fludarabine/cyclophosphamide) in 492 HL patients receiving either a MRD or MUD transplant. The authors found that the intensity of the conditioning did not influence disease relapse, NRM or PFS but the combination fludarabina/cyclophosphamide was associated with a higher NRM beyond 11 months from Halo-HSCT. The lack of differences in terms of OS and NRM between a more intense RIC approach with Flu/Mel 140 and the other ones might be a specific feature of HL patients who have a younger median age compared with NHL subjects and may better tolerate more intensive chemotherapy. Within the Haplo-SCT platform, no specific report was focused on comparing different conditioning regimens in HL patients. In a recent collaborative report between our institution and the Paoli-Calmettes Hospital in Marseille [59], we analyzed the outcome of 147 lymphoma patients, of whom 73 with HL received the Baltimora NMAC regimen and found an encouraging low rate of 1-year NRM (12%) and a 2-year OS and PFS of 77% and 65%, respectively.

To summarize, strategies aimed at improving disease control (by enhancing pre-transplant response, using PBSC as graft source, improving conditioning regimen when active disease is present) and reducing transplant related complications such as aGVHD and infections (by choosing the best available donor or a less toxic conditioning regimen and by adopting better strategies to prevent infections and GVHD) are still debated and need to be explore by future randomized clinical trials. Achievement of these targets may result in a significant enhancement of the main outcomes such as OS, PFS and NRM in patients receiving Haplo-SCT.

## 6. Results of Haplo-SCT in NHL

A few retrospective studies analyzed the outcomes of NHL patients after Haplo-SCT and performed a comparison with other donor sources. Relative to UD transplants, Kanate et al. [27] performed a sub analysis within the aforementioned study and reported similar OS and PFS between the two platforms for the 4 main NHL histologies comprising follicular (FL), diffuse large B cell (DLBCL), mantle cell (MCL) and peripheral T cell (PTCL) lymphomas. In a retrospective analysis by EBMT, Dietrich et al. [60] analyzed 59 NHL patients receiving a Haplo-SCT and compared their outcomes with those of patients transplanted from a MRD (*n* = 2024) or MUD (*n* = 437) during the same time interval. Similar to HL retrospective studies, no differences were identified in terms of OS, PFS, NRM and RI (Table 1). Moreover, aGVHD and cGVHD rates were similar among different donor sources with a tendency for a lower frequency of extensive cGVHD after Haplo-SCT compared with other transplant types. Of note, 2-year OS, PFS, RI and NRM after Haplo-SCT were in agreement with those reported for MRD transplants: 56%, 50%, 27% and 23% respectively (Table 4). Garciaz et al. [61] retrospectively analyzed the outcome of 79 NHL patients receiving either Haplo-SCT (*n* = 26) vs. UD (*n* = 28) or MRD (*n* = 25) transplants and described similar rates of 2-year OS (77% vs. 71% vs. 83%), PFS (65% vs. 68% vs. 80%) and GRFS (65% vs. 54% vs. 66%) among the 3 donor sources (Table 4).

Several studies, both single-center and registry based, have focused on the impact of Haplo-SCT with PT-Cy on different NHL histologies summarized in Table 4 [60,61,62,63,64,65,66].

Regarding DLBCL subtype, Kanate et al. [27] showed again similar outcomes between Haplo-SCT and UD transplants in terms of 3-year OS and PFS. In particular, 3-year OS after Haplo-SCT were 58% and 44%, respectively (Table 4). Dreger et al. [62] performed a large retrospective analysis for the CIBMTR comparing the outcomes of 132 Haplo-SCT with 525 MRD and 781 MUD transplant for patients with R/R DLBCL. All patients received a NMA/RIC regimen and most Haplo-SCT recipients had a TBI-based conditioning followed by BM cells as graft source. Consistent with previous retrospective studies on unspecified lymphoma populations, 3-year OS, PFS, NRM and RI, after Haplo-SCT were comparable with those of other donor types and they were 46%, 38%, 22% and 41%, respectively (Table 4). Moreover, Haplo-SCT was associated with a lower incidence of cGVHD and an improved 2-year GRFS compared with other transplant types. Of note, a subset analysis showed that the benefit in terms of GRFS might be related to the more-prevalent usage of BM instead of PBSC in the Haplo-SCT platform. One of the most relevant observations of this study is that Haplo-SCT was not affected by a higher incidence of disease relapse, despite the very modest anti-lymphoma activity of the conditioning regimen employed and the reduced sensitivity of DLBCL to the GVL effect compared with other NHL subtypes. Therefore, the Haplo-SCT platform may have a stronger GVL effect compensating for the reduced activity of the conditioning regimen and despite the lower incidence of cGVHD.

The main report focused on MCL comes from the sub analysis of the CIBMTR where Kanate et al. [27] again reported similar 3-year OS and PFS between Haplo-SCT (Table 4) and UD transplants with or without ATG: 60% vs. 49% vs. 54% and 51% vs. 33% and 41%, respectively.

A few studies have analyzed the outcome of T cell lymphoma patients receiving Haplo-SCT. Kalankry et al. [63] for the Baltimora group. We recently retrospectively analyzed 29 patients with PTCL receiving a Haplo-SCT with PT-Cy at our institution and at the Paoli Calmettes hospital in Marseille between 2010 and 2019 (manuscript accepted) [66] and compared their outcome with that of 20 MRD and 19 UD transplants in the same time period. Two-year OS, PFS and RI for all 69 patients were 70%, 51% and 21% with no differences according to donor type. In particular, Haplo-SCT recipients had a 2-year OS, PFS, GRFS and RI of 76%, 72%, 59%, 21%, respectively. Haplo-SCT was affected by a significantly lower risk of grade II-IV aGVHD compared with MRD and UD transplants (25% vs. 35% vs. 58%) and with a lower risk of moderate/severe cGVHD (10% vs. 26%) relative to UD transplants. Consistently, receiving a UD transplant was associated with the higher risk of NRM.

To summarize, results from studies focused on different NHL histologies confirm the efficacy and feasibility of Haplo-SCT that should be the transplant of choice when a MRD or MUD is not available, in accordance with the recent position statement of the EBMT [30]. Consistent with findings in HL, Haplo-SCT seems to be characterized by a reduced incidence of cGVHD and improved GRFS compared with MRD or MUD transplants.

## 7. Risk Factors and Unmet Needs of Haplo-SCT in NHL

Risk factors affecting the outcome of NHL patients receiving Haplo-SCT, summarized in Table 5, are similar to the well-known variables identified in the setting of allogeneic transplant from other donor types. The two large studies by Ghosh [28] and Kanate [27] found that both disease related factors, such as pre-transplant disease status (being not in CR), intermediate or high disease-risk index (DRI), histology different from FL, bulky or extranodal disease and patient related factors, such as age, Karnofsky performance status, HCT-CI, were the main independent variables affecting OS, PFS, NRM and RI. These data need to be considered with caution because they originate from a mixed population comprising either haploidentical or MRD or MUD transplants and given the presence of HL and NHL patients, even if NHL represented more than 70% of the subjects. Other reports focused only on patients receiving Haplo-SCT were less powerful to identify risk factors and further analysis based on large number of patients are needed to better explore unmet needs related to this platform. For instance, Kanakry et al. [63] identified timing of remission (first vs. beyond first) and occurrence of GVHD as the only independent predictors of OS and RI, respectively. Results from these studies mainly point out that improving pre-transplant disease control and developing strategies to reduce post-transplant relapse incidence are the most important unmet needs for NHL patients receiving Haplo-SCT.

Better disease control need to be pursued especially for histologies different from FL or for patients at high-risk of relapse. New immunotherapeutic strategies, comprising CAR-T [20], new drug-conjugate antibodies targeting CD19 expressing cells [67] or bispecific T cell engagers [68], represent powerful tools to bring patients with refractory NHL, until now convicted to palliative treatments, to allogeneic transplant. Allogeneic transplant should be reserved to patients not reaching CR with CAR-T, while the other mentioned strategies may be employed as a bridge to transplant given the short median time of response duration.

Role of conditioning regimen for NHL patients is still a matter of debate. In our recent report [59] employing a NMAC regimen for patients with advanced lymphomas, of whom 74 had NHL, resulted in a low 2-year NRM rate (12%) and a promising 2-year OS and PFS of 69% and 65%, respectively. NMAC regimen apparently did not increase the chance of disease relapse (2-year RI: 18%). More recently, Ghosh et al. [69] reported a large retrospective analysis (*n* = 1823 patients) comparing three different RIC-NMA conditioning regimens ranging from a high (fludarabina/melphalan 140 mg/mq) to intermediate (fludarabina-busulfan 6.4 mg/Kg; fludarabina/cyclophosphamide) to lower (fludarabina/cyclophosphamide with TBI 2 Gy) intensity spectrum. Patients had all NHL and transplants were from a UD, either MRD or MUD. Differently from previous CIBMTR [70,71] studies reporting no differences between MAC or RIC-NMA conditioning regimen for R/R NHL, Ghosh et al. [69] found that the more intense Flu-Mel 140 mg/mq was associated with a higher incidence of cGVHD, resulting in a higher NRM rate (26% vs. 17%, *p* < 0.001) and lower OS (49% vs. approximately 60%, *p* < 0.001). This observation is somehow different from a similar comparison in patients with HL [58] and is probably due to the fact that NHL patients are usually older and are less tolerant to more intense conditioning. We recently performed a multicenter prospective phase II clinical trial (manuscript submitted) employing a more intensive RIC regimen, comprising active drugs against lymphoma, such as thiotepa, cyclophosphamide and fludarabina with the aim of reducing RI. We found an interesting low 4-year RI (28%) without any increase in 4-year NRM (15%), resulting in a 4-year OS and PFS of 65% and 54%, respectively (personal observations). While the potential benefits and pitfalls of a myeloablative relative to reduced intensity conditioning regimen before Haplo-SCT have been recently reported [72] in patients with acute leukemia and myelodysplastic syndrome, the role of conditioning regimen intensity in the Haplo-SCT setting needs to be better characterized in lymphomas to address. In particular, we need to address whether it is possible to improve OS by enhancing disease control with more intense conditioning without impacting on NRM or whether it is more advantageous to adopt truly NMA condition to rely on the GVL effect with low incidence of NRM.

Post-transplant strategies comprise prophylactic DLI or usage of new drugs, such as bispecific antibodies or checkpoint blockades that may rescue donor T-cells from tolerance. Up to date the largest published experience on strategies aimed at improving post-transplant disease control for patients with lymphoma concerns DLI [6,7,8,9,10,11,73]. Few studies have been published so fare in the setting of Haplo-SCT either for the treatment of disease relapse or to prevent disease relapse in patients considered at higher risk. Zeidan et al. [74] reported the John Hopkins experience for therapeutic DLI (tDLI) on 40 patients relapsing after Haplo-SCT, of whom 11 had lymphoma. Twelve patients (30%) achieved CR with a median duration of response of 11.8 months. Grade II-IV aGVHD developed in 25% of the subjects and 3 patients had chronic GVHD. The authors suggested that 1 × 10^6 CD3/Kg cells as a safe starting dose. Ghiso et al. [75] reported the Genoa experience on tDLI on 40 patients, of whom 10 with HL where DLI were preceded by 1–3 cycles of chemotherapy. Cumulative incidence of grade II-IV aGVHD was 14% and OS was 52% for the whole population. Results were particularly promising in the HL population with 70% ORR rate (40% CR), 2-year OS of 80% and a low incidence of aGVHD (10%). Cauchois et al. [76] recently reported the experience of our center together with the Paolo-Calmettes hospital of 36 patients, of whom 6 had lymphoma and were receiving prophylactic DLI (pDLI) because they were considered at high risk of relapse. Cumulative incidence of moderate-severe chronic GVHD was 33% with 2 patients dying of cGVHD and one of septic shock in the absence of GVHD resulting into a 9% 1-year NRM. One-year RI was 16% with a promising 1-year PFS of 83% and OS of 76%. Considering the high disease risk of this population, these reported relapse rates and survivals after pDLI are promising, justifying the prospective evaluation of pDLI in further randomized prospective studies. The role of checkpoint inhibitors (CPI) or other immunomodulating strategies is still matter of debate. CPI, such as those harnessing the PD-1/PD-L1 axis, may be highly effective in patients with HL relapsing after allo-HSCT but their effect is jeopardized by significantly increased risk of GVHD-related morbidity and mortality. A recent review of the literature reported aGVHD 14%, cGVHD 9%, ORR rate 54% (33% CR) [47]. Less is known on CPI in disease other than HL. A recent publication [77] retrospectively analyzed the outcome of 21 patients receiving CPI (nivolumab or ipilimumab) for myeloid malignancies or NHL (*n* = 5). Consistently with HL reports, ORR was interesting (43%) but hampered by a high rate of toxicity due to 48% grade II-IV aGVHD and 29% moderate-severe cGVHD. Interestingly, combination of CPI with DLI resulted in a strikingly higher degree of response (80%). Combinations of DLI with bispecific or trispecific T cell engagers have not been published so far, even if phase I/II trial have been announced in the past [78].

A recent consensus form the acute leukemia working party of EBMT [79] has summarized current evidences on tDLI, pre-emptive and pDLI after Haplo-SCT. They concluded that risk of GvHD after unmanipulated DLI in the Haplo-SCT with PT-Cy platform is comparable to an unmanipulated DLI from an HLA-matched donor and that patients with high-risk myeloid malignancies may benefit from a prophylactic haplo-DLI, which should be used in the setting of a clinical trial. Accordingly, clinical trials are needed to explore the efficacy of pDLI for patients with lymphoma after Haplo-SCT and to evaluate potential combination of DLI with other immunotherapeutic modalities.

## 8. Conclusions

Haplo-SCT with PT-Cy as GVHD prophylaxis is a promising platform to cure patients with R/R lymphoma when a HLA identical donor, either matched related or matched unrelated, is not available. Moreover, compared with a MUD transplant, Haplo-SCT has important advantages such as better timing to find a donor and increased chances of post-transplant immune-modulation. Improving disease control and reducing NRM are warranted in order to further improve the outcomes of lymphoma patients.

## Figures and Tables

**Table 1 jcm-09-03589-t001:** Main studies comparing by multivariable analysis the outcomes of hematologic malignancies receiving a MRD, MUD or CBU transplants vs. Haplo-SCT with PT-Cy.

Study	N° Patients (Haplo-SCT)	Comparison	OS (HR, 95% CI)	PFS or DFS (HR, 95% CI)	NRM (HR, 95% CI)	aGVHD (HR, 95% CI)	cGVHD (HR, 95% CI)	RI (HR, 95% CI)	GRFS (HR, 95% CI)
Meybodi 2019	7806 (1410)	Haplo vs. MRD	1.18 (0.92–1.20)	1.03 (0.78–1.38)	**1.36** (**1.12–1.66**)	1.14 (0.82–1.59)	**0.55** (**0.41–0.74**)	0.88 (0.66–1.18)	1.00 (0.50–2.00)
Gagelmann 2019	22,974 *	Haplo vs. MRD	**1.17** (**1.05–1.30**)	-	**1.20** (**1.04–1.40**)	**1.32** (**1.07–1.62**)	**0.46** (**0.33–0.62**)	1.01 (0.86–1.17)	0.87 (0.66–1.15)
Gagelmann 2019	22,974 *	Haplo vs. MUD	1.06 (0.96–1.18)	-	**0.75** (**0.61–0.92**)	**0.76** (**0.62–0.93**)	**0.49** (**0.34–0.71**)	1.20 (1.03–1.40)	**0.69** (**0.52–0.93**)
Gagelmann 2019	22,974 *	Haplo vs. MMUD	**0.79** (**0.65–0.97**)	-	0.51 (0.25–1.02)	**0.51** (**0.32–0.81**)	0.74 (0.54–1.03)	1.06 (0.77–1.47)	0.99 (0.62–1.58)
Kanate 2016	917 (185)	Haplo vs. UD w/o ATG	1.20 (0.90–1.61)	1.11 (0.86–1.40)	0.97 (0.64–1.47)	0.82 (0.57–1.16)	**0.17** (**0.11–0.25**)	1.25 (0.92–1.69)	-
Kanate 2016	917 (185)	Haplo vs. UD with ATG	0.80 (0.59–1.08)	0.86 (0.66–1.13)	0.65 (0.41–1.02)	0.79 (0.52–1.19)	**0.27** (**0.17–0.42**)	1.05 (0.75–1.49)	-
Gosh 2016	987 (180)	Haplo vs. MRD	1.14 (0.87–1.49)	0.98 (0.77–1.23)	1.52 (0.99–2.34)	1.40 (0.99–1.99)	**0.06** (**0.01–0.42**)	0.80 (0.61–1.04)	-
Fatobene 2020	740 (526)	Haplo BM vs. CBU	**0.64** (**0.49–0.84**)	**0.69** (**0.54–0.87**)	**0.52** (**0.37–0.62**)	**0.54** (**0.40–0.73**)	**0.65** (**0.46–0.91**)	0.90 (0.65–1.25)	-
Fatobene 2020	740 (526)	Haplo PBSC vs. CBU	**0.62** (**0.45–0.86**)	**0.53** (**0.40–0.72**)	**0.44** (**0.28–0.68**)	0.96 (0.69–1.31)	0.99 (0.67–1.44)	**0.66** (**0.44–0.97**)	-
Martinez 2017	709 (98)	Haplo vs. MRD	1.24 (0.84–1.82)	0.82 (0.61–1.12)	1.38 (0.79–2.39)	-	-	**0.69** (**0.48–0.99**)	-
Ahmed 2019	596 (139)	Haplo vs. MRD	1.07 (0.79–1.45)	0.86 (0.68–1.10)	**1.65** (**0.99–2.77**)	**1.73** (**1.16–2.59**)	**0.45** (**0.32–0.64**)	**0.74** (**0.56–0.97**)	-
Dietrich 2016	2520 (59)	Haplo vs. MRD	0.98 (0.64–1.50)	0.73 (0.48–1.12)	0.86 (0.47–1.57)	-	-	1.04 (0.66–1.63)	
Gauthier 2018	151 (60)	Haplo vs. MRD	-	-	-	-	-	-	**0.33** (**0.19–0.58**)

Table legend: Haplo-SCT: haploidentical stem cell transplantation; MRD: matched related donor transplant; MUD: matched unrelated donor transplant; MMUD: mismatched unrelated donor transplant; UD: unrelated donor transplant; CBU: cord blood unit transplant; BM: bone marrow; PBSC: peripheral blood stem cells; HR: hazard ratio; OS: overall survival; PFS: progression free survival; DFS: disease free survival; NRM: non-relapse mortality; aGVHD: grade IIi–IV acute graft-versus-host-disease, cGVHD: chronic graft-versus-host-disease either mild-moderate-severe or moderate-severe according to different publications; RI: relapse incidence; GRFS: GVHD/relapse-free survival; * total number of patients, n° of Haplo-SCT varies according to explored outcome. Bold indicates statistically significant differences between graft sources.

**Table 2 jcm-09-03589-t002:** Results from main studies employing the Haplo-SCT with PT-Cy platform for patients with R/R HL.

Study	N° Patients	OS	PFS	NRM	RI	aGVHD	cGVHD	GRFS
Burroughs 2008	90	58	51	9	40	43	35	-
Raiola 2013	25	77	63	4	31	24	9	-
Kanate 2016	46	68	45	-	-	-	-	-
Gayoso 2016	43	58	48	21	24	39	19	-
Castagna 2017	62	63	59	20	-	23	16	-
Martinez 2017	98	67	43	17	39	33	26	40
Gauthier 2018	151	75	66	9	25	28	15	57
Mariotti 2018	64	54	44	26	13	29	3	47
Ahmed 2019	139	63	38	11	32	45	23	-
Marani 2019	41	76	44	7.5	55	21	12	39
Mariotti 2019	91	67	58	22	20	24	7	52

Table legend: OS: overall survival; PFS: progression free survival; NRM: non-relapse mortality; aGVHD: grade II-IV acute graft-versus-host-disease, cGVHD: chronic graft-versus-host-disease either mild-moderate-severe or moderate-severe according to different publications; RI: relapse incidence; GRFS: GVHD/relapse-free survival.

**Table 3 jcm-09-03589-t003:** Risk factors affecting the outcome of Halo-SCT with PT-Cy in HL patients.

Study	OS	PFS	RI	NRM	GRFS
Raiola 2013	-	Disease Status	-	-	-
Gayoso 2016	Disease Status	Disease Status	Disease Status	-	-
Castagna 2017	Disease Status HCT-CI > 0 PBSC	Disease Status	Disease Status	Disease Status HCT-CI > 0	-
Martinez 2017	Age ≥ 40 KPS Refractory Dis	Age ≥ 40 KPS Refractory Dis	KPS Refractory Dis	Age ≥ 40 PS Refractory Dis	-
Gauthier 2018	-	-	Disease status PET status	-	Disease status
Mariotti 2018	Disease Status	Disease Status	Disease Status	RIC/MAC > NMA	Disease Status NMA > RIC/MAC
Ahmed 2019	Age ≥ 50 PS Disease status	KPS Disease status	KPS Disease status	KPS HCT-CI ≥ 3	-
Marani 2019	HCT-CI ≥ 3	HCT-CI ≥ 3 PET (DS ≥ 4)	HCT-CI ≥ 3 PET (DS ≥ 4)	HCT-CI ≥ 3	HCT-CI ≥ 3 PET (DS ≥ 4)
Mariotti 2019	Disease status PBSC > BM HCT-CI ≥ 3	Disease status PBSC > BM HCT-CI ≥ 3	Disease status	-	Disease status PBSC > BM HCT-CI ≥ 3

Table legend: OS: overall survival; PFS: progression free survival; NRM: non-relapse mortality; RI: relapse incidence; GRFS: GVHD/relapse-free survival; HCT-CI: hematopoietic cell transplant-comorbidity index; KPS: karnofsky performance status; Dis: disease; PS: performance status; DS: Deuville score; PBSC: peripheral blood stem cell; BM: bone marrow.

**Table 4 jcm-09-03589-t004:** Results from main studies employing Haplo-SCT with PT-Cy in patients with NHL.

Study	N° Patients	NHL Subtype	OS	PFS	NRM	RI	aGVHD	cGVHD	GRFS
Kanakry 2013	44	PTCL	43	40	9	39	-	-	-
Kanakry 2015	69	B-cell	73	63	10	20	45	13	-
Garciaz 2015	26	Indol/Agg B o T NHL	77	65	15	19	-	15	-
Zoellner 2015	16	Aggressive B o T NHL	69	50	19	36	37	25	-
Dietrich 2016	59	Indol/Agg B o T NHL	56	50	23	27	-	-	-
Kanate 2016	129	FL	70	66	-	-	-	-	-
Kanate 2016	93	DLBCL	58	44	-	-	-	-	-
Kanate 2016	82	MCL	60	51	-	-	-	-	-
Kanate 2016	79	PTCL	36	32	-	-	-	-	-
Dreger 2019	132	DLBCL	46	38	22	41	34	18	36
Castagna 2020	29	PTCL	76	72	7	21	24	10	59

Table legend: OS: overall survival; PFS: progression free survival; NRM: non-relapse mortality; RI: relapse incidence; aGVHD: grade II-IV acute graft-versus-host-disease, cGVHD: chronic graft-versus-host-disease either mild-moderate-severe or moderate-severe according to different publications; GRFS: GVHD/relapse-free survival; PTCL: peripheral T cell lymphomas, NHL: non-Hodgkin lymphoma, DLBCL. Diffuse large B cell lymphoma; Indol/Agg: indolent and aggressive NHL; FL: follicular lymphoma, MCL: mantle cell lymphoma.

**Table 5 jcm-09-03589-t005:** Risk factors affecting the outcome of Halo-SCT with PT-Cy in NHL patients.

Study	OS	PFS	RI	NRM
Kanakry 2013	First vs. beyond first remission	-	aGVHD/cGVHD	
Kanate 2016	Disease status Age ≥ 60 Not FL DRI Int/High	Disease status Age ≥ 60 Not FL DRI Int/High	Disease status Not FL Extranodal Disease DRI Int/High	Age > 40 KPFS < 90 HCT-CI ≥ 3
Ghosh 2016	Disease status Not FL DRI Int/High Bulky Disease	Disease status Not FL DRI Int/High Extranodal/Bulky Disease	Disease status Not FL DRI Int/High Extranodal/Bulky Disease	KPFS < 90 HCT-CI ≥ 3
Dietrich 2016	Prior Auto-Tx Aging	-	-	Prior Auto-Tx Aging
Dreger 2019	Disease status KPS < 90 HCT-CI ≥ 3 Age ≥ 60	Disease status KPS < 90	Disease status KPS < 90 Time from diagnosis ≥ 24 mm	HCT-CI ≥ 3 Age ≥ 60
Castagna 2020	Older age	-	-	-

Table legend: OS: overall survival; PFS: progression free survival; NRM: non-relapse mortality; RI: relapse incidence; HCT-CI: hematopoietic cell transplant-comorbidity index; KPS: karnofsky performance status; FL: follicular lymphoma; DRI: disease risk index; aGVHD/cGVHD: acute and chronic graft-versus-host-disease.

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
