# Peer review of "Haploidentical Stem Cell Transplantation in Lymphomas—Expectations and Pitfalls"

_jcm, 2020, doi:10.3390/jcm9113589_

Round 1

Reviewer 1 Report

Thank you very much for the opportunity to review the paper 

 Haploidentical Stem Cell Transplantation in 2 Lymphomas: expectations and pitfalls. 

My major remark refers to the structure of the review. I would propose to keep as  leading subjects: 1. a comparison of haplo-SCT to other type of transplants summarizing the results from different reports and 2. The results of haplo-SCT in selected types of lymphomas: HL, DLBCL, FL, MCL compared to other type of transplants and novel therapies such as CAR-T cells keeping the same indices of assessment: i.e. OS, PFS  NRM and GRFS. Each fragment should end with the recommendation when haplo-SCT should be performed for a given type of lymphoma. 

Alternatively,  since the review mainly refers to HL I would limit the review to the role of haploidentical transplants in the treatment of HL and changed the title accordingly.

Specifically:

In the introduction I am not convinced why allo-HSCT remains the most powerful curative option for lymphoma. In addition, it is not clear how curative option of allo-SCT is assessed- OS ? PFS? incidence of CR ? quality of life ?

I suggest to change the paragraph "what is the role of Haplo-SCT". Instead of  listing the results of published reports I would attempt to summarize the results of Haplo-SCT in comparison to MRD transplants, MUD transplants  MMUD and cord blood  based on data from all reports. Such an approach will help the reader to better understand the role of haplo-SCT.

The same remark applies to the fragment what is the role haplo -SCT in lymphomas.After this fragment I would consequently describe the results of haplo-SCT for different type of lymphomas in the same schema.

Are the enough data suggesting that haploidentical transplants should be offered to HL patients without the search for MUD? I am not convinced and such an approach should be verified in a prospective clinical trial.

I do not understand why information about one lymphoma subtype (HL) is provided in three different paragraphs whereas all other subtypes of lymphomas are squeezed to two.

Are any limitations of haplo-SCT?- they should be summarized in a separate paragraph

Minor

Spelling error: exploring t_h_e_ _r_ _o_f_ _r_e_c_i_p_i_e_n_t_s_’ _F_c receptor (line 265)

Author Response

Dear Dr Hill,

We wish to thank you yourself and the Reviewers for the review of our manuscript. As suggested, we submit a revision of the manuscript. All comments by the reviewers have been very well taken and they have been addressed in the following changes:

Reviewer 1

My major remark refers to the structure of the review. I would propose to keep as leading subjects: 1. a comparison of haplo-SCT to other type of transplants summarizing the results from different reports and 2. The results of haplo-SCT in selected types of lymphomas: HL, DLBCL, FL, MCL compared to other type of transplants and novel therapies such as CAR-T cells keeping the same indices of assessment: i.e. OS, PFS  NRM and GRFS. Each fragment should end with the recommendation when haplo-SCT should be performed for a given type of lymphoma. Alternatively, since the review mainly refers to HL I would limit the review to the role of haploidentical transplants in the treatment of HL and changed the title accordingly.

Specifically:

In the introduction I am not convinced why allo-HSCT remains the most powerful curative option for lymphoma. In addition, it is not clear how curative option of allo-SCT is assessed- OS ? PFS? incidence of CR ? quality of life ?

We thank the reviewer for this important comment. We modified the text in order make it more clear why allogeneic transplantation represents the most powerful option for R/R lymphoma, i.e. being curative by achieving long-term control of the disease in terms of OS and PFS. Please see the text at lines 42-45.

I suggest to change the paragraph "what is the role of Haplo-SCT". Instead of listing the results of published reports I would attempt to summarize the results of Haplo-SCT in comparison to MRD transplants, MUD transplants MMUD and cord blood  based on data from all reports. Such an approach will help the reader to better understand the role of haplo-SCT.

We thank the reviewer for this comment and we agree that the paragraph was difficult to understand as a list of studies. As recommended, we summarized the evidences comparing Haplo-SCT with the other donor types (lines 71-88).

The same remark applies to the fragment what is the role haplo -SCT in lymphomas. After this fragment I would consequently describe the results of haplo-SCT for different type of lymphomas in the same schema.

We thank the reviewer again for this comment. Following the reviewer’s comments we have now described the results for different types of lymphoma in the following order: all lymphomas, HL, all NHL, FL, DLBCL, MCL and PTCL (lines 105-123; lines 196-228; lines 322-338 and 340-361). Moreover we described the results following the recommended order: OS, PFS, NRM, etc.

Are the enough data suggesting that haploidentical transplants should be offered to HL patients without the search for MUD? I am not convinced and such an approach should be verified in a prospective clinical trial.

We thank the reviewer for this important point. We believe that there is enough evidence to recommend a haploidentical donor for HL patients without the need of searching for a MUD. However, we prefer to propose a more cautious opinion: confirm EBMT recommendations (MRD->MUD->Haplo) and suggest a prospective clinical trial given the higher risk of NRM with Haplo-SCT.

I do not understand why information about one lymphoma subtype (HL) is provided in three different paragraphs whereas all other subtypes of lymphomas are squeezed to two.

We agree with the reviewer’s comments and we rebalanced the information provided for HL and NHL. Moreover we added further information to Table IV concerning NHL subtypes. In any case, more reports are focused on HL, while unfortunately reports on NHL usually comprise many different subtypes with small numbers for each one, making more difficult to extrapolate results for single NHL subtypes.

Are any limitations of haplo-SCT?- they should be summarized in a separate paragraph

Yes, there are limitations for Halpo-SCT that are represented by different issues such as high rate of NRM (mainly due to infections and acute GVHD as remarked on lines 81-85 and 240-45), choice of the best conditioning regimen, of the best graft source (PBSC vs BM), of the best donor, of the best drug to improve disease control before transplant (all summarized in the paragraphs on risk factors and unmet needs for HL and NHL). While the first issue (NRM due to infections or aGVHD) represents a complication common to all types of donors’ transplants, the other ones are more specific for HL or NHL. Therefore, we maintained the structure of the review with the two paragraphs focused on specific unmet needs in lymphoma patients.

Reviewer 2 Report

Dear Author,

            I have read carefully your manuscript entitled, ‘Haploidentical stem cell transplantation in lymphomas: expectations and pitfalls (ID: jcm-965166).

            The manuscript reviewed the issue of haploidentical stem cell transplantation in different settings and lymphomas with different histology.

            This treatment is a novel platform and therapeutic option in those patients with relapsing lymphomas. The revision is quite notable.

            The manuscript is well written and structure in clinical answers, which help to follow up on the article.

            I did not find any relevant issues or handicaps in the current format of the manuscript. The authors might check some minor typo mistaken (e.g. the first line in the abstract section n=165).

Sincerely

Author Response

Dear Dr Hill,

We wish to thank you yourself and the Reviewers for the review of our manuscript. As suggested, we submit a revision of the manuscript. All comments by the reviewers have been very well taken and they have been addressed in the following changes:

Reviewer 2

I have read carefully your manuscript entitled, ‘Haploidentical stem cell transplantation in lymphomas: expectations and pitfalls (ID: jcm-965166). The manuscript reviewed the issue of haploidentical stem cell transplantation in different settings and lymphomas with different histology. This treatment is a novel platform and therapeutic option in those patients with relapsing lymphomas. The revision is quite notable. The manuscript is well written and structure in clinical answers, which help to follow up on the article. I did not find any relevant issues or handicaps in the current format of the manuscript. The authors might check some minor typo mistaken (e.g. the first line in the abstract section n=165).

We thank the reviewer for these comments. We modified the minor typo mistaken as recommended, for example the one mentioned in the abstract section (line 9)

Reviewer 3 Report

The article deals with a very important topic, especially due to the increasing use of haploident donors. The use of CAR-T in the treatment of lymphomas is also discussed, a broad picture of currently used therapies. The article is well written, the literature is extensive. Some not easy terms make it difficult to read the article, for non-native English speakers. Therefore, I recommend that you make a slight linguistic correction.

Author Response

Dear Dr Hill,

We wish to thank you yourself and the Reviewers for the review of our manuscript. As suggested, we submit a revision of the manuscript. All comments by the reviewers have been very well taken and they have been addressed in the following changes:

Reviewer 3

The article deals with a very important topic, especially due to the increasing use of haploident donors. The use of CAR-T in the treatment of lymphomas is also discussed, a broad picture of currently used therapies. The article is well written, the literature is extensive. Some not easy terms make it difficult to read the article, for non-native English speakers. Therefore, I recommend that you make a slight linguistic correction.

We thank the reviewer for the comments. We performed linguistic corrections together with restructure of the manuscript, as suggested by reviewer 1, in order to make the review more easily understandable.

Reviewer 4 Report

Very nice extensive updodate review  that I enjoyed a lot reading

Author Response

Dear Dr Hill,

We wish to thank you yourself and the Reviewers for the review of our manuscript. As suggested, we submit a revision of the manuscript. All comments by the reviewers have been very well taken and they have been addressed in the following changes:

Reviewer 4

Very nice extensive up-to-date review that I enjoyed a lot reading.

We are very happy to know that the reviewer enjoyed the manuscript and we are very thankful for his appreciation.